# Development of Daily and Extreme Temperature Estimation Model for Building Structures Based on Raw Meteorological Data

Jianyu Yang [1], Yongda Yang [2], Jiaming Zou [1] and Weijun Yang [1,*]

1   School of Civil Engineering, Changsha University of Science & Technology, Changsha 410114, China
2   Hunan Construction Investment Group Co., Ltd., Changsha 410018, China
*   Correspondence: yyyaozhijian@163.com

**Abstract:** For building environments, meteorological factors such as daily mean temperature, extreme temperature and seasonal temperature changes, are essential, as they impact building structures significantly. Due to the importance of detailed and accurate temperature data, and taking Beijing, China, as an example, this paper developed a fast and effective interpolation method to extract hourly meteorological data, based on 30 years' raw meteorological data. With the interpolated data, this paper defined the extreme weather for buildings. Moreover, a temperature model based on probability and statistical analysis was constructed, and the general climate standard for days and extreme climate for typical days with different return periods were obtained. Furthermore, meteorological models for standard annual temperature were also achieved, reflecting the daily variation and annual variation of temperature, and can provide continuous-numerical-simulation parameters for analyzing daily and annual temperature. According to the daily temperature difference obeys the Gumble Distribution, the daily temperature difference in different return periods and extreme climates is obtained by analysis. Therefore, annual temperature ranges of different recurrence intervals and extreme climate are also achieved, and the annual temperature ranges can be used to analyze the effect of different recurrence intervals and extreme weather on building structures.

**Keywords:** cubic spline function; meteorological model; extreme weather; daily temperature deference; recurrence interval





## 1. Introduction

Global climate change the important topic across the world, and it significantly affects the occurrence of extreme climate [1]. In recent years, the frequency and intensity of extreme climate events has increased significantly [1–3], represented by the worsened and prolonged droughts in subtropical regions, and increased surface temperature and annual precipitation [4,5]. The frequent extreme climate events caused by global climate change raise new challenges for building structures [6,7]. The effect of disastrous weather on building structures can be divided into direct effect (for example, wind damage, flood damage, snow disaster) and indirect effect, (for example, extreme temperature, drought). The direct effect refers to the concentrated force or distributed force directly applied to the structure. On the other hand, with the indirect effect, the extreme temperature and drought lead to drastic changes in environmental temperature and humidity, thereby causing damage to the building structures [8–12]. At present, regulations of building structure designs treat only the seasonal temperature effect as an indirect effect [13]. The criteria of regulations are presented in [13]: for steel structures sensitive to temperature changes, the effect of daily mean temperature and daily temperature variations needs to be considered, but the criterion does not provide a specific valuation method. Moreover, relatively little research has been conducted on the indirect effect of extreme climate on building structure. Vivian Meløysund [14] study the influence of snow load and wind load

on the reliability of existing building structures in Norway due to shorter winters and higher winter temperatures. Moreover, the effects of freeze–thaw cycle changes caused by climate change on the built structures were studied by Grossi C M [9]. In the present building structure design code, there is inadequate consideration of the dynamic change of climate, and the designed building structure is difficult to adopt, due to the increasingly harsh extreme climate. To analyze the indirect effect of climate in order to reduce the damage of extreme temperature and daily temperature, and to enhance the safety, applicability, and durability of building structures [10–12], it is necessary to construct the model of daily extreme temperature and mean temperature. In this paper, the original meteorological data recorded from national meteorological stations in Beijing from 1980 to 2010 were used to study the above, and for the analysis.

## 2. Hourly Meteorological Data Analysis Based on Cubic Spline Function Method (FFT)

### 2.1. Selection of Original Meteorological Data and Spline Function Method

The data used in this study were recorded from national meteorological stations in Beijing, including the last 30 years of original meteorology data. The dry-bulb temperature of Beijing meteorological stations, directly related to the extreme weather, is employed as the main parameter to build the meteorological model of daily mean temperature and extreme temperature. Meteorological data were measured four times per day(02:00, 08:00, 14:00, 20:00 Beijing Time) at the Beijing station, a national basic weather station. To build the meteorological model, the observation data was adopted to generate the daily hourly data.

For generating daily hourly data, interpolation methods, such as the polynomial function, can be employed to examine the fitting function expression. The polynomial function is relatively simple and convenient for numerical calculations and theoretical analyses. Furthermore, cubic spline interpolation is a means of describing a smooth curve to solve tridiagonal equations through a series of data points. The curve of the cubic spline interpolation function is relatively smooth, and the interpolation points are continuous. For this reason, the accuracy of a cubic spline interpolation function is high, and it approximates to the primitive function.

The theory of fast Fourier transform is: first, the appropriate window function is chosen to suppress the leakage of long range. Then, in line with the form of the window function, the leakage of short range is amended, by using the interpolation method. The FFT algorithm can reduce the computational complexity and improve the efficiency of calculation, and can offer a relatively simple computing method for estimating amplitude, frequency and phase. Moreover, the accuracy of the data cycle which is estimated by sequential harmonic parameters is higher. Therefore, the FFT method is used widely [15–18].

In this paper, in line with the original time-domain sampled sequence, a new equal interval sequence was reconstructed, using cubic spline interpolation. The reconstructed original meteorological data sequence was processed using the FFT, and the hourly meteorological data was estimated.

### 2.2. The Principle and Implementation of Cubic Spline Function FFT Method
2.2.1. The Principle of Cubic Spline Function

The method of three turning angles, the three-moment method and the B-spline basis function are usually employed to calculate the cubic spline function. These methods determine the interpolation function by estimating unknown parameters and minimizing the unknown parameters by using known conditions. The three-moment method is adopted in this paper. Assume $s''(x_i) = M_i, i = 0, 1, \cdots, n, s(x)$ is cubic polynomial in each sub interval $[x_{i-1}, x_i]$. Therefore, $s''(x)$ is linear function in interval $[x_{i-1}, x_i]$, and it can be presented as:

$$s''(x) = M_{i-1} \frac{x_i - x}{h_{i-1}} + M_i \frac{x - x_{i-1}}{h_{i-1}} \quad i = 1, 2, \ldots, n \quad (1)$$

where

$$h_{i-1} = x_i - x_{i-1}$$

Using the condition $s'(x_i - 0) = s'(x_i + 0)$, $i = 1, 2, \cdots, n-1$, and Equation (2) can be obtained:

$$\mu_i M_{i-1} + 2M_i + \lambda_i M_{i+1} = d_i \ i = 1, 2, \ldots, n-1 \tag{2}$$

where

$$\mu_i = \frac{h_{i-1}}{h_{i-1} + h_i}, \ \lambda_i = 1 - \mu_i$$

$$d_i = \frac{6}{h_{i-1} + h_i} \left( \frac{f_{i+1} - f_i}{h_i} - \frac{f_i - f_{i-1}}{h_{i-1}} \right)$$

$$f_i = s(x_i) \ i = 0, 1, \cdots, n$$

To confirm $s(x)$, the boundary condition need to be supplemented. Moreover, there are three general types of boundary condition:

$$S'(x_0) = f'_0, S'(x_n) = f'_n \tag{3}$$

$$S''(x_0) = f''_0, S''(x_n) = f''_n \tag{4}$$

$$S^{(j)}(x_0) = S^{(j)}(x_n), j = 0, 1, 2 \tag{5}$$

Equation (3) takes the derivative according to $s'(x_i - 0) = s'(x_i + 0), i = 1, 2, \cdots, n-1$, and two equations are obtained:

$$\begin{cases} 2M_0 + M_1 = \frac{6}{h_0} \left( \frac{f_1 - f_0}{h_0} - f'_0 \right) \\ M_{n-1} + 2M_n = \frac{6}{h_{n-1}} \left( f'_n - \frac{f_n - f_{n-1}}{h_{n-1}} \right) \end{cases} \tag{6}$$

Assume:

$$\lambda_0 = 1, \ d_0 = \frac{6}{h_0} \left( \frac{f_1 - f_0}{h_0} - f'_0 \right)$$

$$\mu_n = 1, \ d_n = \frac{6}{h_{n-1}} \left( f'_n - \frac{f_n - f_{n-1}}{h_{n-1}} \right)$$

Equations (2) and (6) can be combined and written as matrix (7):

$$\begin{bmatrix} 2 & \lambda_0 & & & \\ \mu_1 & 2 & \lambda_1 & & \\ & \cdot & \cdot & \cdot & \\ & & \mu_{n-1} & 2 & \lambda_{n-1} \\ & & & \mu_n & 2 \end{bmatrix} \begin{bmatrix} M_0 \\ M_1 \\ \cdot \\ M_{n-1} \\ M_n \end{bmatrix} = \begin{bmatrix} d_0 \\ d_1 \\ \cdot \\ d_{n-1} \\ d_n \end{bmatrix} \tag{7}$$

Equation (8), shown below, can be obtained according to Equation (4):

$$M_0 = f''_0, M_n = f''_n \tag{8}$$

Assume $\lambda_0 = \mu_n = 0, d_0 = 2f''_0, d_n = 2f''_n$, Equations (2) and (8) can also be combined and written as matrix (7).

According to Equation (5), two supplementary conditions can be derived:

$$M_0 = M_n, \lambda_n M_1 + \mu_n M_{n-1} + 2M_n = d_n \tag{9}$$

where

$$\lambda_n = \frac{h_0}{h_{n-1} + h_0}, \ \mu_n = 1 - \lambda_n$$

$$d_n = \frac{6}{h_{n-1} + h_0} \left( \frac{f_1 - f_0}{h_0} - \frac{f_n - f_{n-1}}{h_{n-1}} \right)$$

Equations (2) and (9) can be combined, and written as matrix (10):

$$
\begin{bmatrix}
2 & \lambda_1 & & & \mu_1 \\
\mu_2 & 2 & \lambda_2 & & \\
& . & . & . & \\
& & \mu_{n-1} & 2 & \lambda_{n-1} \\
\lambda_n & & & \mu_n & 2
\end{bmatrix}
\begin{bmatrix}
M_1 \\
M_2 \\
. \\
M_{n-1} \\
M_n
\end{bmatrix}
=
\begin{bmatrix}
d_1 \\
d_2 \\
. \\
d_{n-1} \\
d_n
\end{bmatrix}
\tag{10}
$$

Equations (7) and (10) are the three-moment equations, $M_i, i = 0, 1, \ldots, n$ is the moment of $s(x)$. The coefficient matrix element of this kind of tridiagonal equation is $\lambda_i + \mu_i = 1$, and $\mu_i \geq 0$. As the result, these equations are diagonally dominant, and the $M_i (i = 0, 1, \ldots, n)$ in Equation (7) or Equation (10) can be obtained by using the matrix chase-after method. Then, the results can be obtained.

The above discussion shows that the cubic spline function has the unique solution under the boundary condition.

### 2.2.2. Realization of Cubic Spline Function

The aforementioned cubic spline function was implemented using calculation procedures in Matlab, a commercial program. The meteorological data observed 4 times regularly every day were substituted into the calculation procedure, and the cubic spline function method was employed to obtain the curve which could describe the daily meteorological data change. Finally, the Hanning window FFT algorithm was adopted to analyze and optimize the curve, in order to obtain a smooth and accurate hourly meteorological-data curve.

The steps of the fast calculation of the interpolated FFT algorithm using the cubic spline function are:

(1)  Assuming $n$ interpolation nodes as input, and $a = x_1 < x_2 < \cdots < x_n = b$. The corresponding function values are $f_1, f_2, \cdots, f_n$, the boundary condition is $f_0, f_n$, the desired value is $x_0$.
(2)  Calculating $h_j = x_{j+1} - x_j$ $(j = 1, 2, \cdots, n - 1)$.
(3)  Calculating $\mu_i, \lambda_i, d_i$.
(4)  Calculating $\mu_n, \lambda_0, d_0, d_n$
(5)  Solving Equation (7) or Equation (10) by using the chasing method.
(6)  Outputting the expression of the cubic polynomial in each interval.
(7)  Using the Hanning window FFT algorithm to optimize the cubic spline curve.
(8)  Confirming the closed interval $[x_j, x_{j+1}]$ of $x_0$, and calculatng the interpolation $s(x_0)$.

### 2.3. Hourly Temperature Analysis

The original data for dry-bulb temperature were measured 4 times per day (02:00, 08:00, 14:00, 20:00, Beijing Time), in order to find the daily maximum and daily minimum temperatures. Normally, the daily maximum temperature occurs between 14:00 and 16:00, and the daily minimum temperature occurs before sunrise. In this paper the daily maximum temperature and daily minimum temperature are determined, based on a special meteorological data set of buildings' thermal-environment in China [19]. Together with this, 2:00, 8:00, 14:00, 20:00 and the times of daily maximum temperature and daily minimum temperature are set as the basic interpolation points, to obtain the hourly temperature. Moreover, the method of fast calculation of the interpolated FFT algorithm using the cubic spline function was employed to obtain the hourly temperature.

To verify the rationality of the interpolation method offered by this paper, the 24-hourly measured data from the Miyun station in Beijing and the interpolation method presented in this paper were adopted to obtain the correlation curve of theoretical temperature and measured temperature. The result is shown in Figure 1.

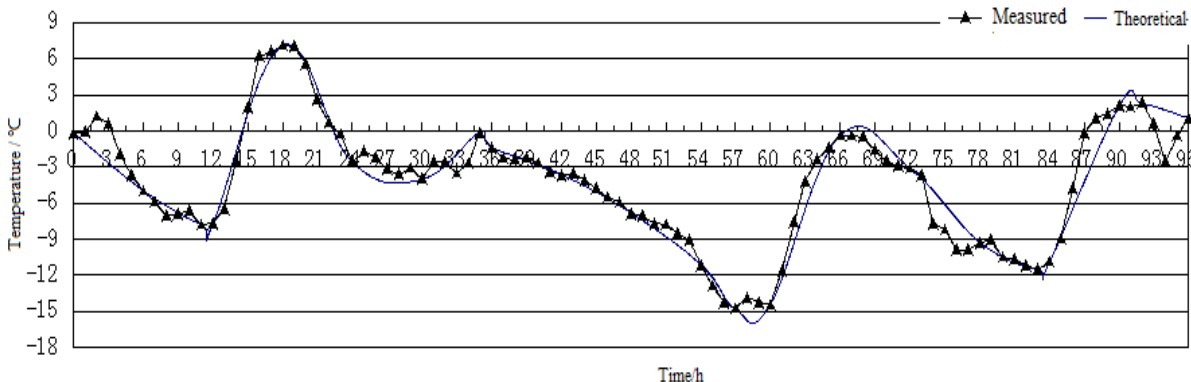

**Figure 1.** Comparison of measured and theoretical temperature of Miyun station in Beijing.

Figure 1 illustrates that the theoretical and measured temperature are the same for 2, 8, 14 and 20 hours, and the maximum temperature and minimum temperature are also consistent. Moreover, the theoretical results conformed well to the measured results, and the fast calculation of the interpolated FFT algorithm using the cubic spline function, performed well.

With this method, the dry-bulb temperatures of the original meteorological data from 1980 to 2010 were obtained and analyzed, and the variation curve of the daily time-continuous meteorological data was calculated. Furthermore, the FFT method was employed to obtain a smoother and more accurate curve of hourly temperature. By averaging the hourly temperature for each day over 30 years, the average hourly temperature for each day in Beijing was obtained, and the annual average hourly temperature for each day in January in Beijing is shown in Figure 2.

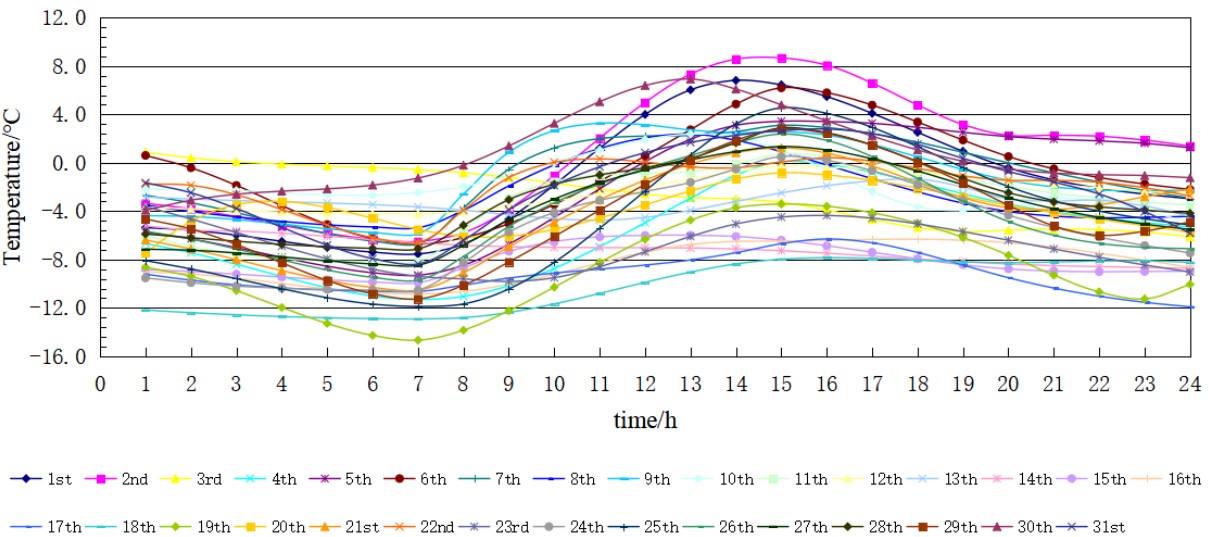

**Figure 2.** Average annual hourly temperature for the days in January in Beijing.

## 3. Establishment of Standard Daily Meteorological Model

The reference temperature is generally the means of the maximum temperature and the minimum temperature, which are obtained by statistical analysis of the years of extreme value of the temperature recorded by the weather station. In *Load code for the design of building structures GB50009-2012* [13], the reference temperature is defined as the monthly average maximum temperature and minimum temperature over 50 years. Moreover, the reference temperature is the standard value of temperature, and the most important meteorological parameter for confirming the temperature effect. The *Fundamental code for*

*design on railway bridge and culvert TB10002.1-2005* [20] adopted the average temperature in July and January as the maximum temperature and minimum temperature.

The monthly average temperature offered by the current standard is not enough to conduct the temperature effect analysis and energy consumption analysis of building structures. As a result, the standard daily meteorological model is put forward to carry out the temperature effect and energy C:\Users\usuario\Users\Jerry\AppData\Local\youdao\DictBeta\Application\7.2.0.0703\resultui\dict\?keyword=consumption C:\Users\usuario\Users\Jerry\AppData\Local\youdao\DictBeta\Application\7.2.0.0703\resultui\dict\?keyword=analysis. To set up the model, the daily hourly temperature in one place and for one month is gathered and treated as the sample value to estimate overall distribution probability. After this, the maximum temperature and minimum temperature over 50 years are found and the curve C:\Users\usuario\Users\Jerry\AppData\Local\youdao\DictBeta\Application\7.2.0.0703\resultui\dict\?keyword=fitting carried out. The continuous temperature-time curve is the standard daily temperature weather model for that month. The temperature effect and energy C:\Users\usuario\Users\Jerry\AppData\Local\youdao\DictBeta\Application\7.2.0.0703\resultui\dict\?keyword=consumption C:\Users\usuario\Users\Jerry\AppData\Local\youdao\DictBeta\Application\7.2.0.0703\resultui\dict\?keyword=analysis of structure can be carried out well with this model. Furthermore, in line with the current standard, the representative maximum and minimum monthly average temperatures are chosen, to analyze the temperature effect and energy C:\Users\usuario\Users\Jerry\AppData\Local\youdao\DictBeta\Application\7.2.0.0703\resultui\dict\?keyword=consumption. Therefore, in this paper, the standard daily meteorological models are set up in July, which has the maximum monthly average temperature and January, which has the minimum monthly average temperature.

### 3.1. Standard Daily Meteorological Model

In order to set up the model, the original observed data are used to solve the hourly meteorological data according to the interpolation method offered in Section 2.2 in this paper, and the annual average hourly temperature for each day in Beijing can be obtained. It is assumed that the temperature at each hour of 24 h per day per month in Beijing follows a normal distribution, which is expressed as $X \sim N(\mu, \sigma^2)$. The distribution C:\Users\usuario\Users\Jerry\AppData\Local\youdao\DictBeta\Application\7.2.0.0703\resultui\dict\?keyword=function is:

$$F(x) = \frac{1}{\sqrt{2\pi}\sigma} \int_{-\infty}^{x} e^{-\frac{(t-\mu)^2}{2\sigma^2}} dt \tag{11}$$

The parameter estimation method is one of the relatively mature methods, and is needed for obtaining the population distribution parameter $\mu$ and $\sigma$ with the sample value. Assume the population $X \sim N(\mu, \sigma^2)$, and $\mu, \sigma$ are unknown C:\Users\usuario\Users\Jerry\AppData\Local\youdao\DictBeta\Application\7.2.0.0703\resultui\dict\?keyword=parameters. Moreover, the point estimation method can also be used for obtaining the population distribution parameters $\mu$ and $\sigma$ with the sample value. The point estimation method estimates the population parameter using the sampling C:\Users\usuario\Users\Jerry\AppData\Local\youdao\DictBeta\Application\7.2.0.0703\resultui\dict\?keyword=statistics. Moreover, the sampling C:\Users\usuario\Users\Jerry\AppData\Local\youdao\DictBeta\Application\7.2.0.0703\resultui\dict\?keyword=statistic is a point value on the number C:\Users\usuario\Users\Jerry\AppData\Local\youdao\DictBeta\Application\7.2.0.0703\resultui\dict\?keyword=axis, so the estimated result is also represented by a numerical value of a point. There are many methods for structuring the estimator $\hat{\theta}(X_1, X_2, X_3, \ldots X_n)$, and the maximum likelihood method is commonly used.

On the premise of a given general distribution, the maximum likelihood method can be adopted. The main idea of this method is: assuming the sample observations of random samples are $x_1, x_{,2}, \cdots, x_n$, to increase the possibility of the sample value, an unknown parameter, $\theta$, is adopted. It can be attributed to the extreme C:\Users\usuario\Users\Jerry\

AppData\Local\youdao\DictBeta\Application\7.2.0.0703\resultui\dict\?keyword=point of the likelihood equation. The solution of the likelihood equation of population C:\Users\ usuario\Users\Jerry\AppData\Local\youdao\DictBeta\Application\7.2.0.0703\resultui\ dict\?keyword=distribution is:

$$\begin{cases} \mu = \frac{1}{n} \sum\limits_{i=1}^{n} x_i = \overline{x} \\ \sigma^2 = \frac{1}{n} \sum\limits_{i=1}^{n} (x_i - \mu)^2 = \frac{1}{n} \sum\limits_{i=1}^{n} (x_i - \overline{x})^2 = B_2 \end{cases} \quad (12)$$

As the result, the maximum likelihood estimations of $\mu$ and $\sigma^2$ are:

$$\begin{cases} \hat{\mu} = \overline{X} \\ \sigma_L^2 = B_2 \end{cases} \quad (13)$$

The maximum likelihood estimation and sample value of the daily 24 hours' temperature of one month are adopted to estimate population distribution and obtain the specific distribution function. The maximum temperature and minimum temperature over 50 years are the estimated by statistical analysis. After this, curve fitting is applied, to fit the average hourly temperature of one month in Beijing. Finally, the curve obtained by fitting the temperature data is employed as the standard daily temperature meteorological model.

The recurrence interval is an estimate of the likelihood of an event. It is a statistical measurement typically based on historic data denoting the average recurrence interval over an extended period of time. The recurrence interval is the numeric equivalent of the reciprocal of event frequency. Therefore, to set up the standard daily meteorological model, assume $F(x) = 1 - 1/R$ and solve the standard maximum temperature $x_R$ when the recurrence interval is R. Moreover, assume $F(x) = 1/R$, and solve the standard minimum temperature $x_R$ as the recurrence interval is $R$. The calculation is shown in Table 1.

**Table 1.** Temperature in July and January in Beijing (°C).

| Month | Hours | Mean Value | Variance | Standard Deviation | 50 Year Return Value | Extreme Weather Value | Month | Hours | Mean Value | Variance | Standard Deviation | 50 year Return Value | Extreme Weather Value |
|---|---|---|---|---|---|---|---|---|---|---|---|---|---|
| | 1 | 25.54 | 3.38 | 1.84 | 29.33 | 29.83 | | 1 | −5.28 | 9.18 | 3.03 | −11.52 | −12.34 |
| | 2 | 24.90 | 3.65 | 1.91 | 28.84 | 29.35 | | 2 | −5.62 | 9.03 | 3.00 | −11.81 | −12.62 |
| | 3 | 24.14 | 4.28 | 2.07 | 28.40 | 28.96 | | 3 | −6.11 | 9.20 | 3.03 | −12.36 | −13.18 |
| | 4 | 23.72 | 4.79 | 2.19 | 28.23 | 28.82 | | 4 | −6.70 | 9.56 | 3.09 | −13.07 | −13.90 |
| | 5 | 23.86 | 4.44 | 2.11 | 28.21 | 28.77 | | 5 | −7.27 | 10.14 | 3.18 | −13.83 | −14.69 |
| | 6 | 24.31 | 3.91 | 1.98 | 28.38 | 28.91 | | 6 | −7.71 | 10.78 | 3.28 | −14.47 | −15.36 |
| | 7 | 24.97 | 3.77 | 1.94 | 28.97 | 29.50 | | 7 | −7.89 | 11.20 | 3.35 | −14.79 | −15.69 |
| | 8 | 25.78 | 4.15 | 2.04 | 29.98 | 30.53 | | 8 | −6.87 | 11.85 | 3.44 | −13.96 | −14.89 |
| | 9 | 26.65 | 4.85 | 2.20 | 31.19 | 31.79 | | 9 | −5.41 | 13.91 | 3.73 | −13.09 | −14.09 |
| | 10 | 27.54 | 5.79 | 2.41 | 32.50 | 33.15 | | 10 | −4.06 | 15.14 | 3.89 | −12.07 | −13.12 |
| | 11 | 28.43 | 6.86 | 2.62 | 33.83 | 34.53 | | 11 | −2.83 | 15.78 | 3.97 | −11.01 | −12.09 |
| July | 12 | 29.28 | 7.73 | 2.78 | 35.01 | 35.76 | January | 12 | −1.74 | 16.69 | 4.09 | −10.16 | −11.26 |
| | 13 | 30.07 | 8.12 | 2.85 | 35.94 | 36.71 | | 13 | −0.82 | 17.87 | 4.23 | −9.53 | −10.67 |
| | 14 | 30.73 | 8.21 | 2.86 | 36.63 | 37.40 | | 14 | −0.12 | 18.51 | 4.30 | −8.98 | −10.14 |
| | 15 | 31.17 | 8.69 | 2.95 | 37.25 | 38.04 | | 15 | 0.22 | 18.48 | 4.30 | −8.64 | −9.80 |
| | 16 | 31.00 | 8.61 | 2.93 | 37.05 | 37.84 | | 16 | −0.09 | 16.87 | 4.11 | −8.55 | −9.66 |
| | 17 | 30.43 | 8.22 | 2.87 | 36.34 | 37.11 | | 17 | −0.78 | 14.71 | 3.84 | −8.68 | −9.71 |
| | 18 | 29.62 | 7.30 | 2.70 | 35.19 | 35.92 | | 18 | −1.66 | 12.49 | 3.53 | −8.94 | −9.90 |
| | 19 | 28.74 | 6.14 | 2.48 | 33.85 | 34.52 | | 19 | −2.59 | 10.66 | 3.26 | −9.32 | −10.20 |
| | 20 | 27.96 | 5.14 | 2.27 | 32.63 | 33.25 | | 20 | −3.39 | 9.72 | 3.12 | −9.81 | −10.66 |
| | 21 | 27.39 | 4.55 | 2.13 | 31.78 | 32.36 | | 21 | −3.96 | 10.01 | 3.16 | −10.48 | −11.33 |
| | 22 | 26.95 | 4.09 | 2.02 | 31.12 | 31.66 | | 22 | −4.39 | 10.49 | 3.24 | −11.06 | −11.94 |
| | 23 | 26.60 | 3.73 | 1.93 | 30.58 | 31.10 | | 23 | −4.75 | 10.50 | 3.24 | −11.42 | −12.30 |
| | 24 | 26.25 | 3.61 | 1.90 | 30.16 | 30.67 | | 24 | −5.05 | 9.89 | 3.15 | −11.53 | −12.38 |

The standard daily temperature meteorological models for Beijing are shown in Figures 3 and 4.

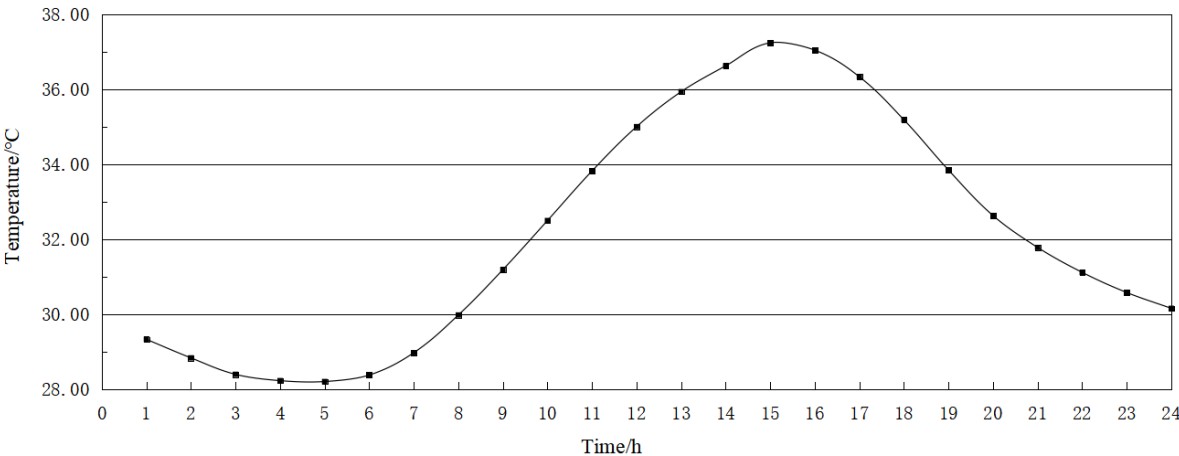

**Figure 3.** Temperature meteorological model of standard day in July in Beijing.

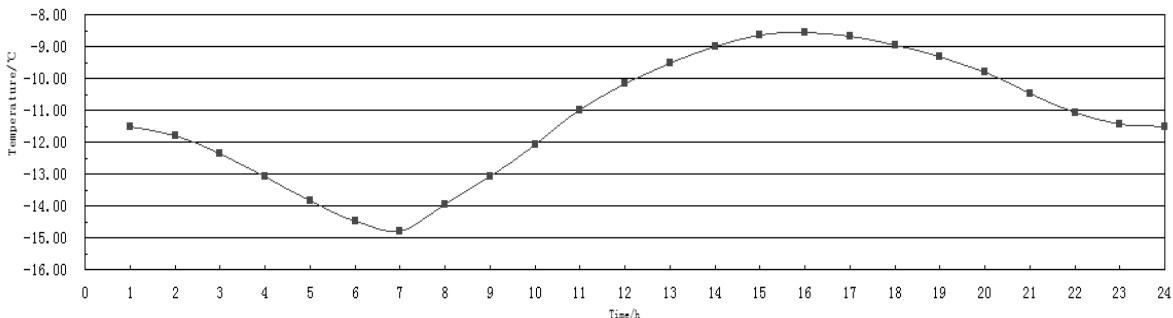

**Figure 4.** Temperature meteorological model of standard day in January in Beijing.

*3.2. Extreme Daily Meteorological Model*

An extreme meteorological climate caused by global warming is becoming more and more frequent. However, the definition of extreme meteorological climate is still unclear in the area of civil engineering. Easterling defines extreme meteorology as a large scale of meteorology abnormality (temperature, rainfall, etc.), or a meteorological event (hurricane, flood, etc.) [21–24]. Moreover, Easterling also considers that the effect of a meteorological event on society can be used to judge extreme meteorology. Generally, however, an extreme meteorological event is different from disastrous weather events. The damage of extreme events can not only can be affected by the intensity of an anomaly of the meteorological climate, but also affected by the area where the disaster occurred and the victims [25–28]. Martin Beniston [29] generalizes three characteristics of extreme meteorological climate: (1) low frequency, (2) high intensity, and (3) huge society loss. The report of the IPCC (Intergovernmental Panel on Climate Change) shows that the possibility of extreme meteorological climate is usually lower than 5% [1,2]. Building structures exposed to natural environments for a long time will be affected by various factors of meteorological environments. But the effect is not as sensitive and direct as the effect of extreme meteorological climate on agriculture and animal husbandry. Moreover, for industrial and civil building construction, the density of population is large and the social economic value is high. Therefore, extreme climate can cause serious social and economic losses. According to the research and the suggestions of the IPCC and Beniston, for civil engineering, extreme meteorological climate is a rare meteorological event whose possibility of occurring is lower than 1% in a geographic area for a certain time.

According to the definition of extreme meteorological, the temperature value with the occurrence probability of 1% (within a 100-year return-period) is the representative daily temperature of extreme weather, and the results of computation are obtained as shown in

Table 1. Moreover, the daily temperature meteorological model of extreme meteorological climate in Beijing is shown in Figures 5 and 6.

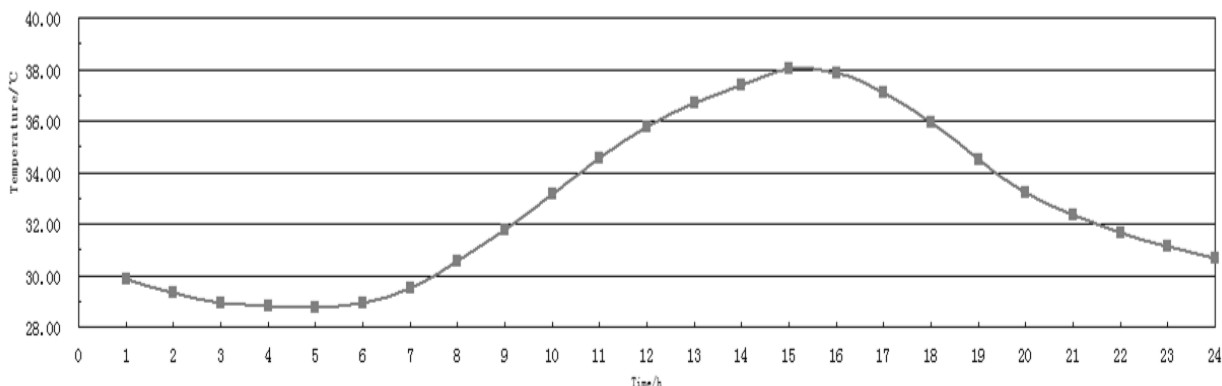

**Figure 5.** Temperature meteorological model of particular day of disastrous weather in July in Beijing.

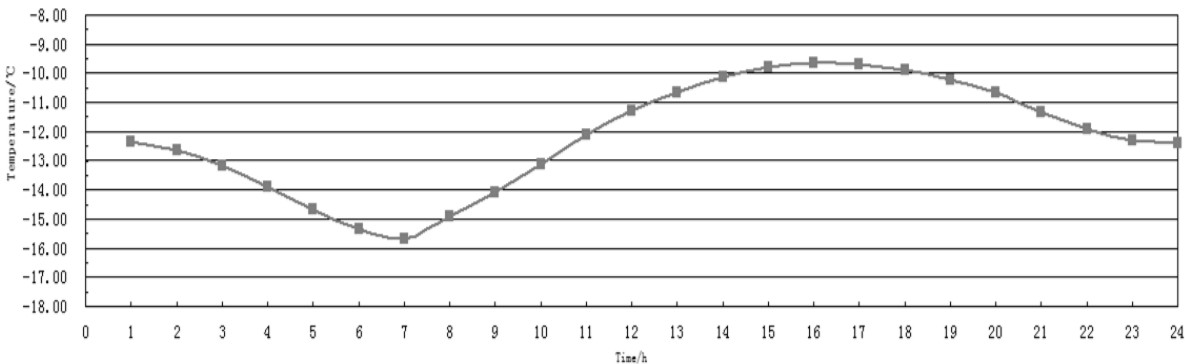

**Figure 6.** Temperature meteorological model of particular day of disastrous weather in January in Beijing.

### 3.3. The Analysis of Daily Temperature Difference

3.3.1. The Analytical C:\Users\usuario\Users\Jerry\AppData\Local\youdao\DictBeta\Application\7.2.0.0703\resultui\dict\?keyword=Method of Daily Temperature Difference

For an ordinary building structure, the temperature value offered by the *Load code for the design of building structures GB50009-2012* [13] is suitable. However, for the structures with fast heat-transfer rate (for example, metal structures and small concrete structures), the standard suggests making corrections to the basic temperature. Currently, such a standard is not offered with a detailed amendment. This problem, though, can be solved through the analysis of daily temperature difference.

The daily temperature difference is treated as a sample. Assuming the sample follows the extreme value type I distribution, the distribution C:\Users\usuario\Users\Jerry\AppData\Local\youdao\DictBeta\Application\7.2.0.0703\resultui\dict\?keyword=function is:

$$F(x) = \exp\{-\exp[-\alpha(x-u)]\} \tag{14}$$

where, $u$ is the mode of distribution, $\alpha$ is the scale parameter of distribution.

When the mean value $\bar{x}$ and standard deviation $\sigma_1$ of finite sample $n$ are treated as the approximate evaluation of $\mu$ and $\sigma$, the distribution parameter $\mu$ and $\sigma$ need to be calculated according to the equation below:

$$\alpha = \frac{C_1}{\sigma_1} u = \bar{x} - \frac{C_2}{\alpha}$$

where $C_1$, $C_2$ can be valued according to the Table E.3.2 in the Load code for the design of building structures GB50009-2012 [13].

Assume $F(x) = 1 - 1/R$, the daily temperature difference $x_R$, of which the recurrence interval is $R$, during the selected month can be calculated by Equation (9):

$$x_R = u - \frac{1}{\alpha} \ln[\ln(\frac{R}{R-1})] \tag{15}$$

3.3.2. The Analysis of Daily Temperature Differences in Different Recurrence Intervals in Different Months

The daily temperature differences of each month in Beijing were gathered, and the mean value $\bar{x}$, standard deviation $\sigma_1$ of the daily temperature difference were calculated afterwards. The results are shown in Table 2. Assuming $R$ equals 10 years, 20 years, 50 years and 100 years, the daily temperature differences were calculated with different recurrence intervals in each month by using Equations (14) and (9). The computations are shown in Table 2.

**Table 2.** Diurnal temperature range each month in Beijing (°C).

| Month | Mean Value | Standard C:\Users\usuario\ Users\Jerry\ AppData\Local\ youdao\DictBeta\ Application\ 7.2.0.0703\ resultui\dict\ ?keyword=Deviation | The Daily Temperature Difference with 10 Years Recurrence Interval | The Daily Temperature Difference with 20 Years Recurrence Interval | The Daily Temperature Difference with 50 Years Recurrence Interval | The Daily Temperature Difference in Extreme Weather |
|---|---|---|---|---|---|---|
| January | 9.2 | 3.6724 | 14.8 | 17.2 | 20.3 | 22.6 |
| February | 11.8 | 2.5364 | 15.7 | 17.4 | 19.5 | 21.1 |
| March | 11.8 | 4.2237 | 18.3 | 21.0 | 24.6 | 27.2 |
| April | 13.0 | 4.1063 | 19.3 | 22.0 | 25.4 | 28.0 |
| May | 12.6 | 4.5861 | 19.6 | 22.6 | 26.4 | 29.3 |
| June | 13.0 | 3.6358 | 18.6 | 20.9 | 23.9 | 26.2 |
| July | 8.3 | 2.6888 | 12.4 | 14.2 | 16.4 | 18.1 |
| August | 9.4 | 1.9732 | 12.4 | 13.7 | 15.4 | 16.6 |
| September | 13.3 | 3.9548 | 19.3 | 21.9 | 25.2 | 27.7 |
| October | 11.5 | 4.0917 | 17.7 | 20.4 | 23.8 | 26.4 |
| November | 11.7 | 3.3483 | 16.8 | 19.0 | 21.8 | 23.9 |
| December | 10.5 | 3.3849 | 15.7 | 17.9 | 20.7 | 22.9 |

According to Table 2, the months with larger daily temperature differences are March, April, May, June, September and October, with an extreme temperature difference of approximately 27.0 °C. Moreover, the months with smaller daily temperature difference are July and August, with an extreme temperature difference of approximately 17.0 °C. The extreme temperature differences of the other months are approximately 22.0 °C.

## 4. The Establishment of Meteorological Model of Standard Year for Buildings

*4.1. The Temperature Meteorological Model of Standard Year*

The standard daily hourly temperatures of each month in Beijing were treated as a sample point, and used to fit the curve. The curve is defined as the temperature meteorological model of a standard year. The model for Beijing is shown in Figure 7.

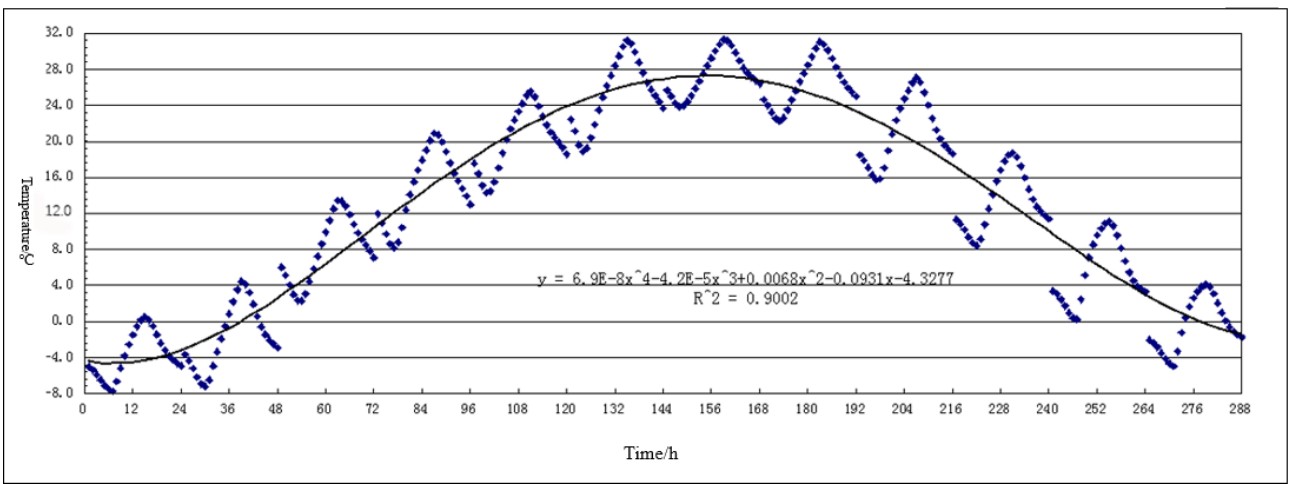

**Figure 7.** Temperature meteorological model of standard year of Beijing.

According to the temperature meteorological model of standard year of Beijing, the fitted curve is shown below:

$$y_T = 6.9 \times 10^{-8}x^4 - 4.2 \times 10^{-5}x^3 + 0.0068x^2 - 0.0931x - 4.33;\ x \in [0, 288] \tag{16}$$

where the coefficient of association $R^2 = 0.9002$.

The temperature-time continuous C:\Users\usuario\Users\Jerry\AppData\Local\youdao\DictBeta\Application\7.2.0.0703\resultui\dict\?keyword=data of a whole year can be obtained by the temperature meteorological model for a standard year. Furthermore, the model reacts to the change in regulation of the annual temperature distribution, and can offer continuous numerical simulation parameters for green building assessment and the energy-consumption evaluation of a building.

### 4.2. Annual Temperature Analysis

According to the definition of annual temperature difference in *Load code for the design of building structures GB50009-2012* [13], the annual temperature difference with different recurrence intervals can be obtained, and the results are shown in Table 3.

**Table 3.** Comparison of standard and calculated annual temperature difference in Beijing.

| Month | | Mean Value | The Reference Temperature with 10 years Recurrence Interval | The Annual Temperature Difference with 10 years Recurrence Interval | The Reference Temperature with 50 years Recurrence Interval | The Annual Temperature Difference with 50 years Recurrence Interval | The Reference Temperature Offered by the Standard | The Annual Temperature Difference Offered by the Standard | The Reference Temperature in Extreme Weather (100 years) | The Annual Temperature Difference in Extreme Weather (100 years) |
|---|---|---|---|---|---|---|---|---|---|---|
| January | maximum | 0.7 | 7.1 | | 13.3 | | — | | 15.9 | |
| | minimum | −8.5 | −11.8 | 47.3 | −13.1 | 52.2 | −13.0 | 49.0 | −13.5 | 54.1 |
| July | maximum | 31.7 | 35.5 | | 39.1 | | 36.0 | | 40.6 | |
| | minimum | 23.4 | 21.1 | | 20.3 | | — | | 20.0 | |

Table 3 illustrates that the annual temperature difference calculated is close to the value offered by the standard. That means the analytical C:\Users\usuario\Users\Jerry\AppData\Local\youdao\DictBeta\Application\7.2.0.0703\resultui\dict\?keyword=method offered by this paper is reasonable. Moreover, the calculated annual temperature difference

is with a 10-, 50-, and 100-year recurrence interval, and it can be used to analyze and research the extreme meteorological environment of building structures.

## 5. Conclusions

This paper develops a method to obtain hourly temperature data by processing meteorological data with the FFT method. With the estimated hourly temperature data, a meteorological model was established for building structures, which offers continuous-numerical-simulation parameters for the analysis of daily temperature effect. Analyses of results suggest several conclusions.

(1) As the daily temperature difference follows the extreme value type I distribution, the daily temperature differences with different recurrence intervals or in extreme weather were obtained by statistical analysis. The results from this paper can contribute to refine the Load code

(2) Code for the design of building structures.

(3) The temperature meteorological model of a standard year reacts to the change in regulation of the annual temperature distribution and offers parameters for the analysis of annual temperature effect.

(4) The annual temperature difference calculated with the method offered by this paper is close to the value offered by the standard. This means the analytical C:\Users\usuario\Users\Jerry\AppData\Local\youdao\DictBeta\Application\7.2.0.0703\resultui\dict\?keyword=method offered by this paper is reasonable. Moreover, the calculated annual temperature difference with different recurrence interval sand in extreme weather can be used for the analysis of building structures in different recurrence intervals or in extreme weather.

**Author Contributions:** Data curation, J.Y. and Y.Y.; Formal analysis, J.Y. and J.Z.; Investigation, Y.Y. and J.Z.; Methodology, Y.Y.; Project administration, W.Y.; Resources, W.Y.; Validation, W.Y.; Writing—original draft, J.Y. All authors have read and agreed to the published version of the manuscript.

**Funding:** This research was funded by The National Natural Science Foundation of China (52209155) and Hunan Water Conservancy Science and Technology Fund (XSKJ2022068-24).

**Institutional Review Board Statement:** Not applicable.

**Informed Consent Statement:** Not applicable.

**Data Availability Statement:** The data that support the findings of this study are available from the corresponding author, [author initials], upon reasonable request.

**Conflicts of Interest:** The authors declare no conflict of interest.

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
