# Peer review of "Development of Daily and Extreme Temperature Estimation Model for Building Structures Based on Raw Meteorological Data"

_applsci, doi:10.3390/app122211582_

Round 1

Author Response

Comment 1:Page 1: While the indirect and direct effects make sense, the way they are described could be slightly improved. The direct effects are characterized as damages (e.g., wind damage, flood damage) while the indirect effects are listed as types of weather events (e.g., extreme temperature, drought). I think the differentiation between these effects could be more clearly defined, ensuring that they are both either entirely impacts or events, instead of their current mix.The rest of the more detailed context explaining direct vs. indirect here makes sense, but this edit would help them be framed more consistently.

Response 1:The classification of disastrous weather effects in this paper refered to other article. Making the differentiation between these effects more clearly will be better. In the future research, I will try to classify various impacts more clearly.

Comment 2:Page 1-2: The current state of work on this related to temperature and the gap you are attempting to fill became a little hard to follow. Are you saying that currently, building design regulations only consider temperature as an indirect effect, such as through how steel is impacted by temperature changes. However, little research has been done on this as it relates to climate change? And to fix this, we need models of daily extreme and mean temperatures to analyze the impact of climate change? Is this summary correct? If so, some editing could be done to smooth these paragraphs.

Response 2: The current code points out that the influence of daily temperature and daily temperature difference should be considered for steel structures sensitive to temperature changes, but no specific value method is given. At the same time, relatively little research has been conducted on the indirect effect of extreme climate on building structure. As th result, I try to construct the model of daily extreme temperature and mean temperature. In the article, I made adjustments in terms of expression.

Comment 3:Page 2, section 2.1:

o What is “original meteorology”?

o What dates and years are included in this analysis?

o Can you provide more detail as to how the dry-bulb temperature from the weather

stations is “directly related to the extreme weather”? How are you relating it?

o You mention using 270 met stations, but then discuss the 1 station in Beijing. Were both

sets of data used? If so, how were they used differently and why did you need both?

o You discuss 4 measurements a day. Is there additional information on what specific times

of day measurements were taken? Some information on this is provided later in the

paper, but should be provided earlier in the paper. This can be an important variable

specifically for extreme temperatures. How were those measurements then used to

create hourly data? Is there a previously established scientific basis for using the FFT

algorithm for this specific purpose and has it been validated against observed data?

Further, if this method is used to interpolate hourly data, instead of saying “hourly

meteorological data is obtained”, it would be more appropriate to say that “hourly

meteorological data was estimated” since you are not directly obtaining and using hourly

measurements.

Response 3:

o“original meteorology” is “original meteorology data”, I have made changes in the text.

oThese data are taken from 1980 to 2010 and are described later in the paper.

o“dry-bulb temperature” describes the temperature in the air. This paper discusses the influence of temperature, not humidity. So, thermodynamic wet-bulb temperature was not adopt.

o I counted 270 weather stations, but the data used in this paper are only from the basic meteorological stations in Beijing. I have made changes in the text.

o The specific times of day are 02:00, 08:00, 14:00 and 20:00 in Beijing time. I have made changes in the text.

Comment 4: Page 4, section 2.2:

Following on the above point, you describe “smooth and accurate hourly meteorological

data were obtained”. Were the data that resulted from this estimation method validated

again actual hourly observations?

Response 4: What I want to express here is that I intend to use this method to obtain smooth and accurate hourly meteorological data curve. In the article, I made some adjustments in expression. In the following article, I have verified the accuracy of this method.

Comment 5: Page 4, section 2.3:

o Similar comment above regarding when these 4 measurements were taken

o Were the maximum and minimum temperatures obtained the daily maximum and

minimum? Or were they the maximum and minimum of just the 4 daily observations?

o The description of “detailed moment of…” and the following discussion of determining the

max/min temperature is confusing and could be reworded. Do you mean the exact time of

the maximum and minimum temperature observations?

Response 5:

The original data of dry-bulb temperature were measured 4 times per day(02:00, 08:00, 14:00, 20:00 Beijing Time), the maximum and minimum temperatures were. The maximum and minimum temperatures are daily maximum and minimum. I made an overall adjustment in this section

Comment 6:Page 5, section 3:

What is the “basic temperature”? How can it be both a maximum and minimum

temperature as well as an annual temperature? Could more distinct scientific terms be

used here? Average and basic do not have the same meaning in the context of

meteorological conditions.

Response 6:

“basic temperature” has been modified to “reference temperature”. the reference temperature is defined as the monthly average maximum temperature and minimum temperature in 50 years. The “reference temperature” can be divided into “reference maximum temperature Tmax” and “reference minimum temperature Tmin”. I have made changes in the text.

Comment 7:Page 7, section 3:

o It would be helpful for the figures on page 7 to have the same y-axis so it was easier to

distinguish the differences between the sets of plots

o I think the phrase “extreme meteorological” is missing a word to most accurately describe

what is increasing in frequency.

o Some of the context as to what you are using to define extreme weather or climate

events and their impact on buildings would be useful in the introduction.

Response 7:

I made appropriate adjustments in the text.

Comment 8:What is “code” annual temperature difference? What is Table 3 comparing the calculated

values to? I didn’t see any specific temperature differences discussed in relation to

“codes” in a way that provided data points with which to compare the calculated values.

Response 8: The“code” is changed to “standard”, the calculated values are compared with the value from standard.

Reviewer 2 Report

The paper discusses about a method to obtain temperature data (hourly and extreme) to be used in the design of buildings. The analyses are performed considering a population of temperature data collected in the area of Beijing as case study.

The topic of the paper is interesting and deserves to be disseminated. Nevertheless, some modifications are required to improve the quality of the paper:

1.       In the Abstract, lines 15-20, the English language needs to be improved; moreover, please use a more formal language (the use of “we constructed…” should be avoided).

2.       In the Abstract, the sentence “As the daily temperature difference follows the I extreme value distribution, the daily temperature difference of different recurrence interval and extreme climate are analyzed and acquired”. The meaning of this sentence is not clear; please explain better the concept.

3.       In the Introduction, Authors discussed about direct and indirect effects on buildings due to climate changes. Even if they recall some references, a brief description of the most common (and probably drastic) effects on buildings should be discussed, mainly to make the reader understand the importance of the topic. Moreover, the citations proposed by Authors are not so actual, so please add some more recent references. Finally, a brief summary of the paper target and contents must be added in this section (Authors spent very few words at lines 81-84, but they are not clear and enough), and the fact that Authors consider climate data of Beijing area as a support for their analyses needs to be declared also in this Section.

4.       In Tab. 1, does the column “Moment” stand for the hours of the day? If yes, please change “Moment” with hours, otherwise define what is means.

5.       Section 4.1, Authors discussed about the temperature meteorological model of a standard year, but in Fig. 7 they reported data (and curve fitting) only considering 12 days. Some comments about that must be added. Moreover, why the daily hourly temperature of Fig. 7 are not continuous? It seems that in the transition between two consecutive days there is a drop in temperature.

In addition, the reviewer has other additional comments about minor typos:

1.       In the Introduction, line 47, the discussed ref. [14] is not consistent with the reference list. Please, correct this typo.

2.       The acronymous FFT must be declared the first time at line 74.

3.       The Eq. (3) is declared twice, consequently the number of equations must be re-arranged.

4.       In the final paper, the quality of Fig. 2 and Fig. 7 must be improved.

5.       Lines 185-186, 187-188, 239-241, 335-337, the English must be revised.

Author Response

Comment 1:In the Abstract, lines 15-20, the English language needs to be improved; moreover, please use a more formal language (the use of “we constructed…” should be avoided).

Response 1:  I have made changes in the text. The content has been modified to”Moreover, a temperature model based on probability and statistical analysis is constructed, and the general climate standard days and extreme climate typical days with different return periods are obtained.”

Comment 2:In the Abstract, the sentence “As the daily temperature difference follows the I extreme value distribution, the daily temperature difference of different recurrence interval and extreme climate are analyzed and acquired”. The meaning of this sentence is not clear; please explain better the concept.

Response 2: I have made changes in the text. The content has been modified to”According to the daily temperature difference obeys the I extreme value distribution, the daily temperature difference in different return periods and extreme climates is obtained by analysis.”

Comment 3:  In the Introduction, Authors discussed about direct and indirect effects on buildings due to climate changes. Even if they recall some references, a brief description of the most common (and probably drastic) effects on buildings should be discussed, mainly to make the reader understand the importance of the topic. Moreover, the citations proposed by Authors are not so actual, so please add some more recent references. Finally, a brief summary of the paper target and contents must be added in this section (Authors spent very few words at lines 81-84, but they are not clear and enough), and the fact that Authors consider climate data of Beijing area as a support for their analyses needs to be declared also in this Section.

Response 3: In the article, I made appropriate adjustments.

Comment 4: In Tab. 1, does the column “Moment” stand for the hours of the day? If yes, please change “Moment” with hours, otherwise define what is means.

Response 4: I have made changes in the text.

Comment 5:Section 4.1, Authors discussed about the temperature meteorological model of a standard year, but in Fig. 7 they reported data (and curve fitting) only considering 12 days. Some comments about that must be added. Moreover, why the daily hourly temperature of Fig. 7 are not continuous? It seems that in the transition between two consecutive days there is a drop in temperature.

Response 5: In Fig.7,The standard daily hourly temperatures of each month in Beijing were treated as sample point and used to fit the curve. So, the reported data considering 12 mounths. I have made changes in Fig.7.

Comment 6:In the Introduction, line 47, the discussed ref. [14] is not consistent with the reference list. Please, correct this typo

Response 6: I have already adjusted.

Comment 7:The acronymous FFT must be declared the first time at line 74.

Response 7: I have already adjusted.

Comment 8: The Eq. (3) is declared twice, consequently the number of equations must be re-arranged.

Response 8: I don't quite understand what experts mean. can you tell me where the Eq.(3) declared twice.

Comment 9:  In the final paper, the quality of Fig. 2 and Fig. 7 must be improved.

Response 9: I have already adjusted.

Comment 10: Lines 185-186, 187-188, 239-241, 335-337, the English must be revised.

Response 10:I have already adjusted.
